# Investigating Ecological Momentary Assessed Physical Activity and Core Executive Functions in 18- to 24-Year-Old Undergraduate Students

**DOI:** 10.3390/ijerph20206944

**Published:** 2023-10-19

**Authors:** Ayva-Mae Gilmour, Mhairi J. MacDonald, Ashley Cox, Stuart J. Fairclough, Richard Tyler

**Affiliations:** 1Movement Behaviours, Health, and Wellbeing Research Group, Department of Sport and Physical Activity, Edge Hill University, Ormskirk L39 4QP, UK; mhairi.macdonald@edgehill.ac.uk (M.J.M.); faircls@edgehill.ac.uk (S.J.F.); 2Division of Musculoskeletal and Dermatological Sciences, Faculty of Biology, Medicine and Health, The University of Manchester, Stopford Building, Oxford Road, Manchester M13 9PT, UK; ashley.cox@manchester.ac.uk

**Keywords:** physical activity, executive function, working memory, inhibition, cognitive flexibility, ecological momentary assessment, guidelines, Pathverse

## Abstract

Although evidence for young children (<10) and older adults (>64) highlights an association between physical activity (PA) and executive functions (EFs), there is a paucity of research on adolescents aged 18–24 years. Thus, this study examined the associations between PA and EF and the difference in EF between individuals who achieve the moderate-to-vigorous (MVPA) guidelines and those who do not. Forty-seven participants engaged in a Stroop task, a reverse Corsi-block test, and a task-switching test, to measure inhibition, working memory, and cognitive flexibility, respectively. An ecological momentary assessment (EMA) was used to determine the participant’s MVPA and step count, through the “Pathverse” app. Multiple regressions were run to predict the task-switch cost, the Stroop effect, and the backward Corsi span from time spent in MVPA. A two-way ANCOVA examined the effects of achieving the MVPA guidelines on EF. MVPA and step count did not significantly predict EF. There were no significant differences in EF between participants achieving the MVPA guidelines and those that did not. Time spent in MVPA and step count were not significantly associated with working memory, cognitive flexibility, or inhibition in adolescents. Further research is warranted to understand other factors that may significantly affect EF, within and outside an individual’s control.

## 1. Introduction

Physical activity (PA) is any bodily movement produced by skeletal muscle that results in energy expenditure [1]. More broadly, PA is promoted through individuals’ preferences, feelings, and ideas via movement and performance within specific cultural contexts [2]. The WHO [3] has established that adults aged 18–64 years should engage in a minimum of 150–300 min of moderate or 75–150 min of vigorous PA per week. Adolescence can be understood as the time frame between childhood and adulthood, relating to individuals aged 10–24 years [4]. In adolescents, engaging in PA and achieving the guidelines is associated with physiological benefits [5], such as a decreased likelihood of developing type 2 diabetes, obesity, and heart disease [6]. There is also evidence for a positive association between PA and mental health [7]; however, a smaller emphasis is placed on the association between PA and cognition or cognitive benefits in adolescents [4,8]. Therefore, research into the effects of PA on cognition is warranted.

Ecological momentary assessment (EMA) studies aim to capture changes in momentary behavior and experiences by capturing data multiple times [9]. Ecological momentary assessment encompasses various methods to obtain real-time data within a real-world setting [10]. Ecological momentary assessment potentially alleviates the limitations of autobiographical memory, whereby an individual recalls their past experiences, as research surrounding autobiographical memory highlights how unreliable memory can be [11]. Throughout EMA, an individual’s natural environment is emphasised to gather the ecological element of behaviors (such as PA), enhancing the ecological validity, through generalisation to the individual’s everyday life [10]. To assess PA behaviors, EMA focuses on an individual’s current behavior, rather than recall [10]. This eliminates any margin for error and bias associated with retrospection, highlighting the momentary element of EMA [10]. Moreover, EMA considers time and intraindividual variability, which may generate different perspectives on presenting PA behaviors [12]. Therefore, using EMA to collect real-time PA data over retrospective methods is favored. 

Within cognition, executive function (EF) is an umbrella term [13] for mental operations involving focus and attention, to control our thoughts and behaviors, especially under situations that are out of the ordinary [14,15,16]. Core EF incorporates neural functions to control individual behaviors and produce preferred results [13]. The frontal lobes of the brain are imperative in measuring an individual’s cognition, through three core EFs: working memory, cognitive flexibility, and inhibitory control [8,17]. Memory stores information when tasked with a mental activity [18]. This process comprises replacing unnecessary data with relevant data, modifying retained information, and programming new information that pertains to the task [19]. Inhibitory control alludes to the capacity to suppress automatic reactions when necessary [15]. Cognitive flexibility enables individuals to adjust their behavior to fit into the environment [20] and cognitively withdraw from activities, plan a response, and apply it to a task [21]. Therefore, advanced EF can enhance academic ability, develop teamwork and leadership expertise, and provide more efficient stress responses and greater organisation [22].

The current literature on adolescents involves top-class sport athletes (known as elite sport), which requires remarkable physiological, cognitive, and perceptive skills [23]. Within sports, individuals must process constant information within a limited time, while under psychological pressure [23]. The mental construct of “perceptual-cognitive skills” alludes to the ability to indent and recognize information concerning their environment [24]. This information is merged with pre-existing knowledge, which enables the selection and execution of responses [23]. Elite athletes are found to perform better on processing speed and attentional measures [25]. Therefore, those within elite sport demonstrate greater EF abilities, and so the outcome of the sport is positively influenced [23,25]. It may be of benefit for coaches to integrate cognitive testing as a tool to optimize athletic development [23]. The comparative literature surrounding the EF abilities of athletes and non-athletes highlights more efficient EF performance in those elite performers [25]. More specifically, non-elite athletes have been found to perform poorly in tests of memory, attention, and decision-making skills [25]. Although sport and EF research places a focus on elite athletes and the influence of elite sport on EF [16,23,25,26], very little is known about whether an association exists in non-athletes, as well as between PA levels, meeting the PA guidelines, and EF. Thus, this study places a focus on PA and PA guidelines in hopes of widening this field. 

Although evidence of a positive association between PA and EF exists [27,28,29,30,31], the literature heavily focuses on children and older adults, so a deficit remains for adolescents aged 18–24 years [32]. Within the UK adult population, only one study on university students (with a mean age of 19 years) has investigated this association and found that increased levels of MVPA were associated with greater task-switching performance [33], which reiterates the potential benefits that PA may have on cognitions [34].

In addition, EMA within PA research is scarce [12], especially for adolescent studies. Thus, the current study aimed to (1) investigate the associations between EMA-derived PA and the core EFs (working memory, inhibitory control, and cognitive flexibility) of adolescents and (2) examine the difference in EF between individuals who meet the recommended PA guidelines and those who do not.

## 2. Materials and Methods

### 2.1. Participants and Settings

This cross-sectional study in northwest England used convenience and snowball sampling techniques to recruit 47 participants (76.6% females; age 20.1 ± 1.7 years). Convenience sampling involves the researcher announcing the study and participants self-selecting if they choose to participate [35]. Snowball sampling enables participants to refer new potential participants to the researcher [35]. Both sampling methods were a form of non-probability sampling and were used as an efficient method to gain participants [35]. The sampling time frame was November 2022 to February 2023. The participants were required to be an undergraduate student at university and aged between 18 and 24 years. The study excluded individuals who could not be physically active or had conditions impacting their memory or color-blindness. Ethical approval was granted by the Sport and Physical Activity Department’s Research Ethics Committee at Edge Hill University (SPA-REC-2022-093) before any research was undertaken. All participants provided informed written and verbal consent before starting the study. 

### 2.2. Measures and Procedures 

#### 2.2.1. Physical Activity

Habitual PA was measured for one week through the Pathverse app, version 1.31.0, Canada (https://pathverse.ca/en/) (accessed on 10 October 2022). Pathverse is an online tool that obtained undergraduate students’ PA levels through EMA. The use of the app within the current study was divided into four phases: (1) researcher Pathverse training and design features ideas, (2) formation of the mobile PA study, (3) pilot study of the app, and (4) implementation of the study. This process can be seen in Appendix A.

Data were extracted from the Pathverse app after one week and reviewed to determine whether participants achieved the PA guideline. The logged physical activities included a Borg rating of perceived exertion (RPE) (0–10) [36], a tool to measure the participant’s effort toward an activity, their exertion, and breathlessness [37]. The category-ration scale (CR-10) (0–10) was used to determine the intensity rate of participants’ physical activities, based on their activity RPE. Light PA ranged from 1 to 3, moderate PA from 4 to 6, and vigorous PA was rated 7–10 [38,39]. Grant et al. [40] compared this with various other linear scales, including the Likert scale, and concluded that the reproducibility of the results aligned with but also outperformed some linear scales [37]. The participants’ total amount of moderate physical activity (MPA) and vigorous physical activity (VPA) were calculated and compared with the MVPA guidelines. The guidelines alluded to a minimum of 150–300 min of moderate or 75–150 min of vigorous PA per week [3]. The participants were stratified based upon this criterion: those that achieved the MVPA guideline (group 1) and those that did not achieve the MVPA guideline (group 2).

The participants’ daily step values were obtained via the Pathverse app, through the synching of various fitness apps. Those apps included Apple Health, Google Fit, and Fitbit and tracked the participants’ steps via their phone or fitness watch. The participants were stratified based upon this criterion: those that achieved the step guidelines (group 1) and those that did not achieve the step guidelines (group 2).

#### 2.2.2. Executive Functioning

The participants’ core EFs were measured through a battery of cognitive tests via the Psytoolkit online software (version 3.4.2) [41,42] (https://www.psytoolkit.org/c/3.4.2/survey?s=BFThW) (accessed on 10 October 2022) on desktop computers in an ICT laboratory at Edge Hill University. Before the cognitive tests, an online survey was coded into the study to collate data that acted as covariates due to the possibility of a statistical relationship with the dependent variables. The survey asked participants for their age, sex, and average academic attainment (average grade percentage at university (%)). Home address postcodes were also required to calculate the English index of multiple deprivation (EIMD) deciles [43] to relatively measure deprivation across small areas within England. The EIMD ranks every small area in England from 1 (most deprived) to 32,844 (least deprived), and the deciles are calculated from these [43]. Once completed, the participants proceeded with three cognitive tests that assessed cognitive flexibility, inhibition, and visuospatial working memory. After each test, the participants were required to input their scores into a data sheet.

#### 2.2.3. Cognitive Flexibility

A task-switching test was implemented to assess the participants’ cognitive flexibility [44,45]. This task was used due to its high internal consistency, validity, and good test–retest reliability [46]. This task involved two individual tasks (A and B), in which participants carried out a trial of each and then a trial that was a combination of tasks A and B presented on a grid format. Task A asked participants to respond to a letter when presented next to a number (i.e., A3), and task B required a response to the number rather than the letter. In the combination trial, participants had to respond to the stimuli based on its location within the grid. The less time participants took to complete the task-switch trials, the more proficient their task-switching ability. 

#### 2.2.4. Inhibition

A Stroop task was implemented to assess the participants’ inhibition through a compatible and incompatible trial [47]. For instance, one trial presented the color and meaning of a word to be the same, e.g., the word “green” was in green font (compatible). The other trial displayed a word with a different meaning and color, e.g., “green” was in red font (incompatible) [48]. The task presented the name of a color (e.g., red) but asked participants to identify the font color in which it was written. The participants’ greater performance in the compatible trial indicated a lower level of interference in their reading ability and greater overall performance [48].

#### 2.2.5. Visuospatial Working Memory

A reverse Corsi block test was used to assess the participants’ visuo-spatial working memory because it is a valid and reliable assessment strategy [49]. This task presented nine blocks that illuminated in a sequence. As the trials progressed, the number of illuminated blocks increased. The participants were required to retain the reverse order of the sequence and input this by selecting the squares when prompted. This provided an indication of the participant’s spatial span; the greater the sequence retained, the more efficient their spatial span was.

### 2.3. Data Analysis and Statistical Analyses

Descriptive statistics, the mean and standard deviation, were obtained on all measured variables (Table 1). A two-way analysis of covariance (ANCOVA) was conducted to assess the differences in executive function (cognitive flexibility, inhibition, and visuospatial working memory) between sex and achieving the PA guidelines or not, while controlling for EIMD deciles, age, and academic attainment. The covariates highlighted were selected to eliminate any extraneous variables measurement of EF, given that positive correlations have been shown [27,50,51]. Two multiple regressions investigated the association between EF (cognitive flexibility, inhibition, and visuospatial working memory) and MVPA and between EF and step value, including academic attainment, sex, age, and EIMD decile in each model. This enabled the analysis of the importance of each predictor on the above potential association and determined whether the MVPA and step value predicted increases in EF. 

## 3. Results

A multiple regression was run to predict the task-switch cost, Stroop effect, and backward Corsi span from the MVPA, academic attainment, EIMD decile, sex, and age. Partial regression plots and a plot of studentized residuals against the predicted values identified no linearity. A Durbin–Watson statistic of 2.3 (task-switch cost), 2.2 (Stroop effect), and 2.3 (backward Corsi span) confirmed the independence of the residuals. A plot of studentized residuals versus unstandardized predicted values confirmed homoscedasticity. The assumption of normality was achieved, as assessed by a Q–Q plot. The final model did not statistically predict the task-switch cost, *F* _(5,28)_ = 0.93, *p* = 0.475, R^2^ = 0.14, Stroop effect, *F*
_(5,28)_ = 0.44, *p* = 0.817, R^2^ = 0.07, or backward Corsi span, *F*
_(5,28) =_ 1.87, *p* = 0.133, R^2^ = 0.25. Regression coefficients and standard errors for the final model are displayed in Table 2.

A second multiple regression was run to predict the Stroop effect, task-switch cost, and backward Corsi span from the daily step value, academic attainment, EIMD decile, sex, and age. Partial regression plots and a plot of the studentized residuals against the predicted values identified no linearity. A Durbin–Watson statistic of 2.1 (Stroop effect), 2.2 (task-switch cost), and 2.4 (backward span) confirmed the independence of the residuals. A plot of the studentized residuals versus the unstandardized predicted values confirmed homoscedasticity. The assumption of normality was met, as assessed by a Q–Q plot. The final model did not statistically predict the Stroop effect, *F*
_(5,28)_ = 0.63, *p* = 0.317, R^2^ = 0.10, task-switch cost, *F* _(5,28)_ = 1.03, *p* = 0.394, R^2^ = 0.16, or backward span, *F*
_(5,28)_ = 1.97, *p* = 0.510, R^2^ = 0.26. The regression coefficients and standard errors are displayed in Table 3.

A two-way ANCOVA was conducted to examine the effects of MVPA on EF, after controlling for age, EIMD decile, and academic attainment. There was a linear relationship between the Stroop effect, task-switch cost, and backward Corsi span for each group, as assessed by visual inspection of a scatterplot. There was homogeneity of the regression slopes. The studentized residuals plotted against the predicted values for each group confirmed homoscedasticity, and there was homogeneity of variances as assessed by Levene’s test of homogeneity of variance (*p* = 0.626 backward span, *p* = 0.922 Stroop effect, *p* = 0.957 task-switch). The data had no outliers as there were no cases with studentized residuals greater than ±3 standard deviations. The leverage values and Cook’s distance confirmed no leverage or influential points. As assessed by Shapiro–Wilk’s test (*p* > 0.05), the studentized residuals were normally distributed. 

There was no significant two-way interaction between the Stroop effect *p* = 0.786, backward span *p* = 0.598, and task-switch *p* = 0.915, with achieving the PA guidelines, while controlling for age, EIMD decile, and academic attainment. Therefore, an analysis of the main effects was not performed. The means, adjusted means, standard deviations, and standard errors are presented in Table 4 for the Stroop effect, backward Corsi span, and task-switch.

## 4. Discussion

This study aimed to (1) investigate whether an association existed between PA and the core EFs of adolescents and (2) whether a difference occurred in the core EFs of those who achieved the PA guidelines and those who did not. Moreover, aligning with previous research [27,31,32,33,34,35,36,37,38,39,52], it was hypothesised that individuals who achieved the recommended PA guidelines would obtain greater working memory, inhibitory control, and cognitive flexibility than those who did not achieve the recommended PA guidelines, while an increase in MVPA would be associated with greater EF.

Overall, this study highlighted that a significant association did not exist between visuospatial working memory, inhibition, and cognitive flexibility with MVPA in adolescents aged 18–24 years. There were no significant differences in EF between those who met the PA guidelines and those who did not meet the PA guidelines. This rejected the hypotheses as a greater level of MVPA engagement did not associate with the visuospatial working memory, task-switch, or Stroop effect testing scores. In addition, the participants’ daily step count was also explored in relation to their EF, and it was also found that steps were not significantly associated with greater EF. While the findings were unexpected and rejected the hypotheses, it is important to understand the factors that withheld the potential to explain the above findings.

### 4.1. Associations between Physical Activity and Executive Function 

The literature highlights other studies that failed to demonstrate the potential association between PA and EF within children and older adults [53,54,55]; in line with the current study, Ho, Gooderham, and Handy [56] also failed to establish this association within university students. Several methodological differences exist between this study and the work of Ho, Gooderham, and Handy [56], which add depth to this field since the results align. The first difference alludes to Ho, Gooderham, and Handy’s [56] use of the flanker task, which activates similar brain regions to the Stroop test, such as the anterior cingulate cortex [57]. Moreover, Ho, Gooderham, and Handy [56] utilised the International Physical Activity Questionnaire (IPAQ) long form to measure participants’ PA, whereas this study opted for EMA. Given that the IPAQ can be subject to recall bias, which may provoke an overestimation of PA [58], the use of EMA helped to broaden this field by providing a new light on potentially more accurate PA measures. Thus, this study adds to the current literature via a novel methodological approach. 

Although not a key aim of the study, the ANCOVA results demonstrated no significant differences in PA and EF between males and females within our sample. Thus, males and females were placed into the same group for the multiple regression analyses. Sex-related differentiation has been found to occur within associations between PA and cognition [27,59,60]. Adolescence has been seen to be associated with a decline in PA as age increases [61,62]. It has also been highlighted that adolescent boys undergo a decrease in their PA levels much earlier and obtain a greater level of sedentary behavior than adolescent girls [63]. This may be driven by psychological factors, such as life transitions, i.e., completing mandatory schooling and starting a job [63]. This can also stem from motivational differences and interests [64] and having access to sporting opportunities given that curriculum-based PA ends once individuals leave school [65]. There is evidence to suggest that biological sex has an influence on memory [27], which may be influenced by physiological and psychological factors that can change in response to PA [66]. It has been highlighted that females demonstrate greater cognitive outcomes that are associated with PA [59,60]. For instance, there is evidence to suggest that the impact of acute PA on episodic memory was found to be greater on females than on males [67]. Despite the literature highlighting these interesting findings, this study’s results did not align. Therefore, sex-related differentiation was not found to play a key role in the findings of this study. 

Further, the task-switching test is a measure of latency as opposed to absolute, due to the difference in the mean reaction time between switch and non-switch trials being the measure of task-switching ability [68]. However, this result is inaccurate as switch costs can also occur [68], so considering alternative tests of cognitive flexibility, such as the cognitive flexibility scale [69], may be warranted for future research. However, it should be noted that other research [27] that found an association between PA and cognitive flexibility utilized the trail-making test [70], which analyzes errors and speed combined. This suggests that the task-switching test used in this study shines a new light on cognitive flexibility. Therefore, the way in which cognitive flexibility is measured via cognitive testing should be considered prior to those tests being carried out. 

Moreover, research has demonstrated significant associations in terms of EFs and elite sports. For example, higher EF abilities have been reported from elite athletes when compared with non-athletes [71,72,73], and greater EF has been found in elite athletes when compared with sports performers with less experience or expertise [16,26,74]. Within adolescence, it has been found that elite soccer players obtained greater EF scores than a standardized norm group of males and females [75]. An approach known as the “cognitive component skills approach” investigates the association between sports expertise and cognitive test performance that are relevant to the cognitive requirements in elite sports [76]. Specifically, this approach investigates cognitive functions including working memory, cognitive flexibility, and inhibition [76]. Some studies failed to align with these results [77,78]. Although elite sports have demonstrated significance in terms of bettering an individual’s EF [26,71,72], this factor was not accounted for in this study, as this study placed a focus on non-elite athletes, PA level, and meeting the PA guidelines. Therefore, participants were not questioned whether they participated in sport at an elite level, and thus, the potential association between PA and EF may still exist if the confounding variable was included. 

### 4.2. Exploring the Factors That May Influence Executive Function 

While an association between PA and EF has not always been found [53,56], discussing the potential reasons behind this is imperative to gain a more in-depth understanding of the results obtained. The literature highlights the negative impact of sleep deprivation on an individual’s cognition [79], yet evidence for this association is equivocal [80]. More specifically, the impact of sleep duration and quality on EF performance is highlighted as this was not accounted for in this study. This is important given that slow-wave sleep (deep sleep) benefits the prefrontal cortex [79], which plays a main role in EF [81]. Wilckens et al. [81] assessed this association on a population of similar age to this study and found that longer sleep duration resulted in greater working memory and inhibition. Notably, Wilckens [81] discovered an association between very short and very long periods of sleep with poorer working memory. Although the measures of working memory differed from Wilcken et al.’s [81] study, it should be noted that a Stroop task was also used as a measure of inhibition, and the study concluded that there was a strong association between sleep and inhibition in adolescents. Moreover, this was also highlighted by Anderson et al. [82], who found that “sleepy” participants obtained poorer EF. Opposing this, longitudinal studies have confirmed that obtaining 6–8 h of sleep per night as an adolescent is associated with enhanced EF later in life [83]. Thus, researchers and health professionals should consider sleep duration as a potential contributor to adolescent cognitive functioning. This factor may explain the fact that there were no significant associations between PA and EF in this study. 

Moreover, an individual’s ability to direct their behavior toward achieving a goal is imperative throughout academic tasks [84]. Therefore, it would be reasonable that EF would be related to academic achievement (AA) [84]. Within school-aged individuals, it has been found that poor EF abilities have been associated with lower academic achievement [85,86], while greater EF performance has been associated with higher achievement in reading and mathematics [87,88]. It is imperative to note that research surrounding the association between EF and AA for the population of this study is scarce [89], and thus, very little is known about whether an association exists. Notwithstanding, this study did not objectively measure the participant’s AA and asked participants to note their “average” academic attainment. This allowed social desirability bias to play a part, and so an association between AA and EF may still exist. Further research is warranted to expand this field within this population. 

While this study did not highlight a significant association between PA and cognitive flexibility or inhibition, it is imperative to note that the cognitive testing occurred at scattered times throughout the day. Participants selected a session that best suited their availability, to complete the battery of cognitive tests [41,42] and be enrolled onto the Pathverse app. Although this made the data collection process more efficient, the literature highlights negative impacts of the time of day and cognitive processing [90]. Folkard and Monk [91] highlighted the impact of the time of day on the efficiency of an individual’s working memory and the speed at which they can retrieve information from their long-term memory [92]. This has been explained through circadian arousal changes that stem from body temperature adaptations throughout the day, namely, an increase as the day progresses, which is said to promote optimum performance on complex cognitive processes, such as working memory [90]. Given that participants completed cognitive testing at different times throughout the day, the literature is suggestive of potential inaccuracies within the EF data of this study. Despite this, the evidence highlights that some individuals report feeling most alert in the morning, and others report these feelings in the evening [93]. Thus, the potential inaccuracies of cognitive testing at various times throughout the day may not be as prominent as first thought.

In addition, while previous results demonstrate the benefits of PA on cognitive performance [94,95], executive capacities function parallel to the frontal lobe of the brain [96], so it is unsurprising that the difference in EF between those who meet the PA guidelines and those who do not may be attributed to genetic variation, as opposed to their PA engagement [96,97]. Evidence highlights the genetic significance of working memory, with approximate heritability from 33 to 49% [98,99]. Key transmitters are critical for optimal working memory, namely, excessive or very little dopamine [100] and noradrenaline [96]. The evidence indicates the role of serotonin in inhibition [101], whereby a polymorphism in the serotonin transporter gene prevents serotonin uptake [102]. This implies that genetic variation among participants may play a role in their cognitive abilities, which is unknown to the researcher. However, some studies have failed to demonstrate this association [103,104]. 

Furthermore, the literature highlights associations between sedentary behavior (SB), sitting or lying down behaviors that incur <1.5 METs [105], and poorer EF [106]. Considering this study focused on achieving PA guidelines, individuals who did not achieve the PA guideline included sedentary participants and those who engaged in little PA that failed to achieve the guideline. This is noteworthy as participants in the “not achieved” group may have engaged in SB, potentially negatively impacting their EF scores. Thus, further analysis is warranted to explore this potential association between SB and EF in adolescents.

Although not the primary aim, this study concluded no associations between daily step values and working memory, inhibition, or cognitive flexibility, after adjusting for confounders, which contradicts previous research [107]. This study used Apple Health, Google Fit, and Fitbit to measure step counts synched through Pathverse. Despite popularity, mobile health through wearable devices, such as watches or armbands, is being questioned regarding the validity and reliability of metric data including step count and heart rate [108]. The evidence implies that the daily step value varies across device brands and types [109]. A review conducted by Bunn et al. [110] concluded that Fitbit wearables underestimated the step count and heart rate, which impacted the participant’s energy expenditure value. Although Fitbit obtains high interdevice reliability for steps, Fitbit may only provide accurate values in very few circumstances [110,111]. Despite this, in a systematic review of nine wearable device brands, Apple and Samsung obtained the greatest validity for step count [108]. Thus, some brands may demonstrate more inaccuracies than others, which may have presented false step data in this study. This suggests that the inaccuracy of the step measurement technology may be responsible for no association being concluded between daily step values and EF. 

### 4.3. Strengths

The strengths of this study allude to the use of a novel mobile health app as a mode of EMA to measure PA. Physical activity assessment relies on self-reported data, which provokes recall bias [10]. Ecological momentary assessment therefore aims to reduce this recall bias and strengthens the ecological validity surrounding the research of factors that may impact behavior in real-life settings [10]. This is because EMA collects data on large populations, which a is proximal to the time and location of the behavior occurrence and so reduces the reliance on memory [112]. Thus, EMA may provide new insights into PA measurements given that recall questionnaires may be less effective in discovering phenomena that vary over time [12]. This study also objectively measured EF through the Psytoolkit software (version 3.4.2) [41,42]. This occurred within a controlled environment, with noise and distraction levels kept to an absolute minimum, which was imperative considering the literature highlights greater cognitive performance in silent conditions [113]. A pilot study was conducted to assess feasibility, which provided insights into the study protocol and methodological complexity [114].

### 4.4. Limitations and Future Directions

This study also had several limitations that warrant consideration. The cross-sectional design did not permit cause and effect [115]. Although the population of this study was under-researched, the availability of time to engage in PA may have been a constraint for university students given their academic calendar [27]. During the recruitment process, university students were discouraged by the PA aspect of this study, and it was found that more females opted in than males. This may stem from females obtaining differing perceptions of risk than males [116] and trust being essential in research participation [117]. The final sample of participants was 47, which limits the generalizability of the results to all university students across England [118].

Furthermore, the exclusion criteria of this study failed to consider the neurodevelopmental condition autism. While autism is characterized by a deficit in social interaction ability [119], a primary phenotype of autism is executive dysfunction [120]. The literature commonly discusses how this is largely present in adulthood [121,122]. This is imperative as executive dysfunction presents brain abnormalities, negatively impacting complex information processing [123]. Given that EF improves throughout adolescence [124], in typical children, there is a call for future research to understand cognitive maturation and the differences present when compared with individuals with autism [125]. Thus, this is suggestive of inaccuracies within the EF abilities of those potential participants with autism since this co-variable is unknown within this study. Future research should factor in cognitive impairments when exploring the physical activity–cognition phenomenon. Likewise, it may benefit this field to explore executive dysfunction concerning autism more widely. 

#### Technical Issues Surrounding the Pathverse App

The Pathverse app presented potential issues surrounding missing and/or false data. For instance, the participants potentially did not log their PA engagement or falsely claimed PA engagement, which may be apparent due to social desirability bias [126,127]. However, EMA within PA research somewhat overcomes this as this method involves honest responses within a participant’s natural environment and relies on episodic memories, so the influence of memory bias is reduced [128]. Ecological momentary assessment enables the participants to privately record their PA engagement, which reduces the pressure to provide answers that are socially desirable [128]. Therefore, EMA encourages more authentically representable answers from the participants. Despite the Borg CR10 scale [38] being deemed reliable [37,40], it is subjective, suggesting a potential underlying bias. The participants were not informed of each value of light, moderate, and vigorous PA, due to the Pathverse app study design prohibiting changes to be made to the pre-set survey designs. Thus, the RPE rating may not be entirely accurate, which may influence the categorisation of their PA. It may be beneficial for the Pathverse app to implement a design feature allowing edits to be made to multi-option survey designs to allow for features, such as RPE, to be used more efficiently. Thus, future research is warranted on improving methods of PA measurement within EMA. Likewise, it would be beneficial to further the measurement of PA through mobile health since this field is growing [129].

Furthermore, the step count accuracy may also be questioned due to technological issues with the sync function, participants remembering to sync their steps, and the potential of false data. Given that the participants’ daily step value data were synched via a watch or mobile phone, there was the drawback of mobile phones failing to track the step count if the device was not physically on/with the individual. Many occasions within this study found synching errors with various apps, resulting in missing step count data. Thus, an association between the daily step value and EF may not have been concluded due to an inaccuracy in the step value.

Future research should use a pedometer given that they increase credibility and are continually praised for their accuracy in step measurement [130]. A further consideration should be aimed toward the use of accelerometers, given that they provide the step count, accelerations, time spent in PA intensities, sleep quality and duration, and sedentary behaviors [131]. Pathverse may wish to consider a future feature of enabling synching of pedometer and accelerometry data to the Pathverse app, to allow for greater accuracy and broader data in future research, as well as contextual PA data through EMA.

## 5. Conclusions

This study adds depth to the physical activity–cognition phenomenon, by analysing habitual PA through EMA (diminishing the recall biases presented in other self-report measures), with objectively measured EF on a population that greatly warranted research within this field. The role of meeting the PA guidelines in enhancing adolescents’ EF was not found, and MVPA was not associated with greater working memory, inhibition, or cognitive flexibility. Exploring other potential influences in improving an individual’s EF is needed. Although the findings may not align with research within this field, the role of PA in EF must not be dismissed in future research. Furthermore, a light needed to be placed on adolescents, in the hopes that this research study provokes further analysis on this population to diminish the existing deficit surrounding the role of PA in the EFs of adolescents.

## Figures and Tables

**Table 1 ijerph-20-06944-t001:** Descriptive characteristics of the participants: mean and standard deviation (M(SD)) unless indicated otherwise.

Variables	All	Sex		Physical Activity Guidelines	
		Males	Females	Achieved	Not Achieved
*n*	47	11	36	11	36
Age (years)	20.1 (1.4)	20.2 (1.6)	20.1 (1.3)	19.8 (1.0)	20.3 (1.5)
Females (*n*)	36	-	-	10	26
Males (*n*)	11	-	-	1	10
Academic attainment (%)	64.2 (7.3)	59.9 (6.8)	65.2 (7.1)	63.1 (6.5)	64.6 (7.6)
IMD decile	5.4 (3.2)	5.6 (3.4)	5.3 (3.2)	4.3 (3.4)	5.7 (3.2)
Physical activity					
MVPA (minutes)	78.1 (116.2)	38.6 (58.6)	90.2 (127.0)	255.6 (107.7)	23.9 (38.9)
RPE	3.9 (2.3)	3.3 (7.1)	2.3 (4.1)	4.9 (3.9)	2.0 (6.4)
Step value (number)	7688.5 (3516.8)	8362.4 (4368.7)	7495.9 (3283.6)	8562.3 (2985.7)	7438.8 (3654.8)
Executive function					
Stroop effect (m/s)	74.4 (101.9)	88.7 (93.6)	70.0 (105.2)	68.8 (75.2)	76.1 (109.7)
Corsi-backward span (number of items)	4.5 (2.3)	5.1 (2.1)	4.3 (2.4)	3.8 (2.7)	4.7 (2.2)
Task-switch cost (response time in m/s)	440.4 (296.1)	346.6 (259.0)	366.3 (304.1)	456.2 (261.5)	435.6 (309.2)

**Table 2 ijerph-20-06944-t002:** Multiple regression results for MVPA and Stroop effect, backward Corsi span, and task-switch cost.

Model	*B*	95% CI for *B*		SE *B*	β
		LL	UL		
Stroop Effect					
MVPA	0.09	−0.22	0.40	0.15	0.11
Academic attainment	0.64	−4.73	6.00	2.62	0.05
IMD decile	−4.38	−16.02	7.25	5.68	−0.15
Sex	38.03	−59.96	135.75	47.71	0.16
Age	12.19	−21.99	46.37	16.68	0.15
Backward Corsi Span					
MVPA	−0.01	−0.01	0.00	0.00	−0.24
Academic attainment	0.07	−0.05	0.19	0.06	0.22
IMD decile	0.19	−0.06	0.45	0.12	0.26
Sex	0.35	−1.80	2.48	1.04	0.06
Age	0.21	−0.54	0.96	0.36	0.10
Task-Switch Cost					
MVPA	−0.14	−0.96	0.69	0.40	−0.06
Academic attainment	13.32	−0.88	27.52	6.93	0.38
IMD decile	−13.76	−44.54	17.02	15.03	−0.17
Sex	29.54	−228.99	288.07	126.21	0.05
Age	−29.56	−119.97	60.86	44.14	−0.13

Note. Model = “Enter” method in SPSS Statistics; *B* = unstandardized regression coefficient; CI = confidence interval; LL = lower limit; UL = upper limit; SE *B* = standard error of the coefficient; β = standardized coefficient.

**Table 3 ijerph-20-06944-t003:** Multiple regression results for the step count and Stroop effect, backward Corsi span, and task-switch cost.

Model	*B*	95% CI for *B*		SE *B*	β
		LL	UL		
Stroop Effect					
Step value	0.01	−0.005	0.16	0.005	0.22
Academic Attainment	−0.07	−5.47	5.34	2.64	−0.01
IMD decile	−4.78	−16.08	6.52	5.52	−0.16
Sex	15.56	−83.07	114.19	48.15	0.07
Age	17.91	−17.60	53.42	17.33	0.21
Backward Corsi Span					
Step value	0.00	0.00	0.00	0.00	0.27
Academic Attainment	0.05	−0.07	0.17	0.06	0.17
IMD decile	0.23	−0.02	0.48	0.12	0.31
Sex	0.17	0.88	2.34	1.06	0.03
Age	0.42	0.29	1.20	0.38	0.20
Task-Switch Cost					
Step value	0.01	−0.02	0.04	0.01	0.14
Academic Attainment	12.35	−2.07	26.77	7.04	0.35
IMD decile	−12.62	−42.76	17.51	14.71	−0.15
Sex	10.64	−252.40	273.67	128.41	0.02
Age	−18.31	−113.00	76.39	46.23	−0.08

Note. Model = “Enter” method in SPSS Statistics; *B* = unstandardized regression coefficient; CI = confidence interval; LL = lower limit; UL = upper limit; SE *B* = standard error of the coefficient; β = standardized coefficient.

**Table 4 ijerph-20-06944-t004:** Means and standard deviations (M(SD)), adjusted means and standard errors Madj (SE) for Stroop effect, backward Corsi span, and task-switch for groups.

Females	Physical Activity Guidelines	
Stroop effect		
	Achieved	Not Achieved
M(SD)	69.0 (84.0)	42.0 (104.3)
Madj (SE)	67.0 (72.6)	42.3 (28.8)
Backward Corsi span		
	Achieved	Not Achieved
M(SD)	3.0 (2.7)	5.0 (2.0)
Madj (SE)	3.6 (1.6)	4.6 (0.6)
Task-switch		
	Achieved	Not Achieved
M(SD)	386.0 (260.7)	434.8 (281.1)
Madj (SE)	386.8 (192.4)	412.4 (76.2)
**Males**	**Physical Activity Guidelines**	
Stroop effect		
	Achieved	Not Achieved
M(SD)	n/a	81.4 (95.0)
Madj (SE)	n/a	82.7 (44.0)
Backward Corsi span		
	Achieved	Not Achieved
M(SD)	n/a	4.7 (2.6)
Madj (SE)	n/a	4.9 (1.0)
Task-switch		
	Achieved	Not Achieved
M(SD)	n/a	376.4 (269.8)
Madj (SE)	n/a	439.5 (116.6)

## Data Availability

The corresponding authors are happy to provide data, if required, on reasonable request.

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
