# Peer review of "Investigating Ecological Momentary Assessed Physical Activity and Core Executive Functions in 18- to 24-Year-Old Undergraduate Students"

_ijerph, 2023, doi:10.3390/ijerph20206944_

Round 1

Reviewer 1 Report

Dear Authors 

I am glad I have been offered this opportunity of reviewing your interesting study. Below please find some comments, suggestions and points worth considering to make presentation of your research more valuable. 

Generally, the paper is well-written and of a good quality and tables are neat and easily readable for a reader. 

Title and Abstract are clear, informative enough and connect well with the body of the paper and the present the essence of your study. The only thing that I would suggest reconsidering are Key words - in my opinion there too many and coincide with each other - 'cognition' covers the large part of cognitive/executive functions/flexibility and working memory - maybe it is worth looking at it again.    

To enhance and broaden the quality of your paper I suggested you look at the following issues.  

Introduction - Although this a well-written part of the body text, still it has some limitations and requires enhancement of the rationale for the study. 

You have mentioned the association of PA and cognitive functions in children and elderly, and added that there is little research done in the adult section of the population as an argument for undertaking your study. As much as it is true, looking at your hypotheses I think you should look into the elite sport and cognitive functions research - I suggest referring to papers from Applied Psychology like : "Are expert athletes ‘expert’ in the cognitive laboratory? A meta-analytic review of cognition and sport expertise', or "Measurement of cognitive functions in experts and elite athletes: A meta-analytic review' just to name these two. The mentioned studies also concern adults, but well trained and it would be worthy bringing this issue into the light of a reader. 

Another thing is a sex related differentiation - Is there any ? Have you come across any study, research on the possible differences between male and female adults linked to their PA and cognitive functions? Is it well known that entering adulthood women PA decreases while with young male adults often it is contrary. Is there any association with their emotional (front cortex) and social developments and roles? And if there aren't any this way you can provide arguments why you kept the two sexes (male and female) in one group. 

Methods - generally research procedure and tools are well-defined and described with details and reliability coefficients characteristics. However, some additional information are necessary - for example when you state that the sample was divided into group 1 and group 2 based on fulfilling PA guidelines what recommendations you are referring to? It be better for a reader if one could see those criteria in that section. 

Results - in this section tables are clearly displayed and one can get a sense of findings from the figures, but for example in table 1 there is "Academic attainment' listed as a variable and this was not introduced in the Introduction nor in the Methods section, at least I was a bit surprise when seeing it in the table. Also in the same tables you provide measuring units (minutes,%, n) with some variables but with others - why? 

Another issue is you have two tables numbered as 1 (page 4 and page 5), although in body of the text you refer to table 2. Also there is some typo in table 4 (page 9) in Backward Corsi. 

  In Discussion - since your study did not confirm your pre-assumption it is important to seek for some potential explanation why your hypothesis has been rejected. While bringing sedentary behaviour patterns and sleep deprivation as variables and potential causes seems sensible I would avoid bringing the arguments like students-alcohol in association with EF since you have not collected that kind of data and especially that this may imply some suggestions, but perhaps it is worth digging into academic levels, meeting PA guidelines and EF? and again, bringing the young elite athletes and cognitive functions? 

 Providing Strengths and Limitations of the study is certainly a value and give a reader a positive impression what obstacles you have encountered and what steps need to be taken in the future. 

Conclusions are also good summary of your main findings. 

The whole text is well-referenced, but extending the scope of your Introduction and Discussion will require some more references.  

Good luck with your revision. 

English is fine, no need to any further proofreading

Author Response

Comment #

Comment

Response

1

Generally, the paper is well-written and of a good quality and tables are neat and easily readable for a reader. 

Thank you for these positive comments.

2

Title and Abstract are clear, informative enough and connect well with the body of the paper and the present the essence of your study. The only thing that I would suggest reconsidering are Key words - in my opinion there too many and coincide with each other - 'cognition' covers the large part of cognitive/executive functions/flexibility and working memory - maybe it is worth looking at it again.    

Thank you for this comment. The reasoning behind the use of “cognition” as a key word was that the paper relates to this overall. We also wanted to include the other key words as these relate to the fact the results are separated by each executive function (i.e., working memory, inhibition, cognitive flexibility). Through using these specific key words, we wanted to be clear as to what aspect of cognition this paper entails from the start. Whilst “cognition” covers a large part of this study, the authors believe all other key words in this study are key aspects of this study, and so “cognition” has now been removed from the key words.

3

Introduction - Although this a well-written part of the body text, still it has some limitations and requires enhancement of the rationale for the study. 

You have mentioned the association of PA and cognitive functions in children and elderly, and added that there is little research done in the adult section of the population as an argument for undertaking your study. As much as it is true, looking at your hypotheses I think you should look into the elite sport and cognitive functions research - I suggest referring to papers from Applied Psychology like : "Are expert athletes ‘expert’ in the cognitive laboratory? A meta-analytic review of cognition and sport expertise', or "Measurement of cognitive functions in experts and elite athletes: A meta-analytic review' just to name these two. The mentioned studies also concern adults, but well trained and it would be worthy bringing this issue into the light of a reader. 

Thank you for your comment here, the authors agree and appreciate the literature guidance. The authors have now added to the introduction which can be found in lines 72-89, which reads, “The current literature on adolescents involves top-class sport athletes (known as elite sport), which requires remarkable physiological, cognitive, and perceptive skills [24]. Within sports, individuals must process constant information within limited time, whilst under psychological pressure [24]. The mental construct of ‘perceptual-cognitive skills’ allude to the ability to indent and recognize information concerning their environment [25]. This information is merged with pre-existing knowledge, which enables the selection and execution of responses [24]. Elite athletes are found to perform better on processing speed and attentional measures [26]. Therefore, those within elite sport demonstrate greater EF abilities, and so the outcome of the sport is positively influenced [24,26]. It may be of benefit for coaches to integrate cognitive testing as a tool to optimize athletic development [24] comparative literature surrounding the EF abilities of athletes and non-athletes highlights more efficient EF performance in those elite performers [26]. More specifically, non-elite athletes have been found to perform poorly in tests of memory, attention, and decision-making skills [26]. Although sport and EF research places a focus on elite athletes and the influence of elite sport on EF [24,26–28], very little is known about whether an association exists in non-athletes, as well as, between PA level, meeting the PA guidelines, and EF. Thus, this study places a focus on PA and PA guidelines in hopes of widening this field.”

4

Another thing is a sex related differentiation - Is there any ? Have you come across any study, research on the possible differences between male and female adults linked to their PA and cognitive functions? Is it well known that entering adulthood women PA decreases while with young male adults often it is contrary. Is there any association with their emotional (front cortex) and social developments and roles? And if there aren't any this way you can provide arguments why you kept the two sexes (male and female) in one group. 

The authors agree with the points stated, however the ANCOVA demonstrated no differences in PA and EF between males and females within our sample. Thus, males and females were kept in one group for the multiple regressions.

Nevertheless, sex related differentiation has now been added to the discussion on lines 292-309, which reads:  “Although not a key aim of the study, the ANCOVA results demonstrated no significant differences in PA and EF between males and females within our sample. Thus, males and females were placed into the same group for the multiple regression analyses. Sex related differentiation has been found to occur within associations between PA and cognition [61–63]. Adolescence has been seen to be associated with a decline in PA as age increases [64,65]. It has also been highlighted that adolescent boys undergo a decrease in their PA levels much earlier and obtain a greater level of sedentary behaviour than adolescent girls [66]. This may be driven by psychological factors, such as life transitions, i.e., completing mandatory schooling and starting a job [66]. This can also stem from motivational differences and interests [67], and having access to sporting opportunities given that curriculum-based PA ends once individuals leave school [68]. There is evidence to suggest that biological sex has an influence on memory [63], which may be influenced by physiological and psychological factors that can change in response to PA [69]. It has been highlighted that women demonstrate greater cognitive outcomes that are associated with PA [61,62]. For instance, there is evidence to suggest that the impact of acute PA on episodic memory was found to be greater on women than men [70]. Despite the literature highlighting these interesting findings, this study’s results did not align. Therefore, sex-related differentiation was not found to play a key role in the findings of this study.”

5

Methods - generally research procedure and tools are well-defined and described with details and reliability coefficients characteristics. However, some additional information are necessary - for example when you state that the sample was divided into group 1 and group 2 based on fulfilling PA guidelines what recommendations you are referring to? It be better for a reader if one could see those criteria in that section. 

Thank you for this suggestion. We agree with this suggestion that the criterion upon which participants were based should be noted in this section. We have now provided this detail. Lines 134-139 now reads: “Participants total amount of moderate physical activity (MPA) and vigorous physical activity (VPA) were calculated and compared to the MVPA guidelines. The guidelines allude to a minimum of 150-300 minutes of moderate or 75-150 minutes of vigorous PA per week [31]. The participants were stratified based upon this criterion, those that meet the MVPA guideline (Group 1) and those that do not meet the MVPA guideline (Group 2)”.

6

Results - in this section tables are clearly displayed, and one can get a sense of findings from the figures, but for example in table 1 there is "Academic attainment' listed as a variable and this was not introduced in the Introduction nor in the Methods section, at least I was a bit surprise when seeing it in the table.

Also in the same tables you provide measuring units (minutes,%, n) with some variables but with others - why? 

Thank you for your comment. We would like to clarify that “Academic attainment” was discussed in the methods section of the paper. Specifically, lines 151-152, which reads “The survey asked participants for their age, sex, and their average academic attainment (average grade percentage at university (%))”. In the “Data analysis and statistical analyses” section of the methods, the rationale behind the covariate of academic attainment was discussed. Specifically, lines 191-193: “The covariates highlighted were selected to eliminate any extraneous variables measurement of EF, given that positive correlations have been shown [23,46,47]”. Therefore, the authors believe this was an important variable to include in the descriptives table.

Thank you for pointing this out. Measuring units have now been added to those that require a unit in red text within the manuscript.

7

Another issue is you have two tables numbered as 1 (page 4 and page 5), although in body of the text you refer to table 2. Also there is some typo in table 4 (page 9) in Backward Corsi. 

Thank you - this has now been corrected and the table on page 5 is now labelled as table 2. The typo of Backwards Corsi has now been corrected and is highlighted in red text within the manuscript.

8

In Discussion - since your study did not confirm your pre-assumption it is important to seek for some potential explanation why your hypothesis has been rejected. While bringing sedentary behaviour patterns and sleep deprivation as variables and potential causes seems sensible I would avoid bringing the arguments like students-alcohol in association with EF since you have not collected that kind of data and especially that this may imply some suggestions, but perhaps it is worth digging into academic levels, meeting PA guidelines and EF? and again, bringing the young elite athletes and cognitive functions? 

Thank you for bringing this to the authors’ attention. We agree with the students-alcohol association with EF, and this has now been removed from the manuscript. In addition, the association of elite sport and EF has been brought into the discussion and can be found on lines 321-336, which reads “Moreover, research has demonstrated significant associations in terms of EFs and elite sports. For example, higher EF abilities have been reported from elite athletes when compared to non-athletes [74–76], and greater EF has been found in elite athletes when compared to sports performers with less experience or expertise [27,28,77]. Within adolescence, it has been found that elite soccer players obtained greater EF scores than a standardized norm group of men and women [78]. An approach known as the “cognitive component skills approach” investigates the association between sports expertise and cognitive test performance that are relevant to the cognitive requirements in elite sports [79]. Specifically, this approach investigates cognitive functions including working memory, cognitive flexibility, and inhibition [79]. Some studies failed to align with these results [80,81]. Although elite sports have demonstrated significance in terms of bettering an individual’s EF [28,74,75], this factor was not accounted for in this study as this study placed a focus on non-elite athletes, PA level, and meeting the PA guidelines. Therefore, participants were not questioned whether they participated in sport at an elite level, and thus the potential association between PA and EF may still exist if the confounding variable was included.”

Please also see lines 357-368 which discusses academic attainment and its association with executive function. This reads “Moreover, an individual’s ability to direct their behaviour towards achieving a goal is imperative throughout academic tasks [77]. Therefore, it would be reasonable that EF would be related to academic achievement (AA) [77]. Within school-aged individuals, it has been found that poor EF abilities have been associated with lower academic achievement [78,79], whilst greater EF performance has been associated with higher achievement in reading and mathematics [80,81]. It is imperative to note that research surrounding the association between EF and AA for the population of this study is scarce [82], and thus very little is known about whether an association exists. That being said, this study did not objectively measure participant’s AA, and asked participants to note down their ‘average’ academic attainment. This allowed social desirability bias to play a part, and so an association between AA and EF may still exist. Further research is warranted to expand this particular field on this population”.

9

Providing Strengths and Limitations of the study is certainly a value and give a reader a positive impression what obstacles you have encountered and what steps need to be taken in the future. 

Thank you for highlighting this.

10

Conclusions are also good summary of your main findings. 

Thank you for this positive comment.

11

The whole text is well-referenced, but extending the scope of your Introduction and Discussion will require some more references.  

Thank you for this comment. Please find the additional references included within the manuscript below the table.

Reviewer 2 Report

Dear Authors,

thank you for your interesting research.

My main concerns are:

Please explain: why do you call 18-25years subjects teenagers?, aren't they young adults?

Please provide additional information on the sample method, the target population or sampling frame of the study and the demographics of the participants in the sample description.

Please clarify: in line 96 you write Habitual PA was measured for one week through the Pathverse app, but in supplementary document Phase 4: you write: All participants were provided with detailed instructions on how to use Pathverse and continued to log PA for two weeks.

When selecting the subjects, was it taken into account whether they are athletes or not?

Explain how MET was calculated?

What issues and questions remain unresolved or emerge from the results in this study?

Author Response

Comment #

Comment

Response

1

Please explain: why do you call 18-25years subjects teenagers?, aren't they young adults?

Thank you for highlighting this point. The authors understand this misunderstanding. To ensure a clear understanding of the population of this study, a definition has been added that this study appertains to. This can be found on lines 37-38, which reads “Adolescence can be understood as the timeframe between childhood and adulthood, relating to individuals aged 10-24years [4]”.

2

Please provide additional information on the sample method, the target population or sampling frame of the study and the demographics of the participants in the sample description.

Thank you for this comment. Additional information has now been added regarding the sample methods. Please see lines 105-109, which now reads “Convenience sampling involves the researcher announcing the study and participants self-selecting if they choose to participate [37]. Snowball sampling enables participants to refer new potential participants to the researcher [37]. Both sampling methods were a form of non-probability sampling and were used as an efficient method to gain participants [37]”.

Line 110 now reads “The sampling time frame was November 2022 to February 2023”. Lines 99-100 state that participants were located in Northwest England, and that 76.6% of participants were female.

Lines 110-113 highlight the inclusion and exclusion criteria, which reads “Participants were required to be an undergraduate student at University and aged between 18-25 years. The study excluded individuals who could not be physically active, have conditions impacting their memory, or colour-blindness”. The authors believe this provides sufficient information on the population and the sample within the current study.  

3

Please clarify: in line 96 you write Habitual PA was measured for one week through the Pathverse app, but in supplementary document Phase 4: you write: All participants were provided with detailed instructions on how to use Pathverse and continued to log PA for two weeks.

Thank you for highlighting this. The supplementary material Phase 4 has now been corrected and states “All participants were provided with detailed instructions on how to use Pathverse and continued to log PA for one week”.

4

When selecting the subjects, was it taken into account whether they are athletes or not?

It was not taken into account whether the participants were athletes or not. Thank you for picking this up, this has now been discussed and can be found in lines 321-336 in red text within the manuscript, which reads “Moreover, research has demonstrated significant associations in terms of EFs and elite sports. For example, higher EF abilities have been reported from elite athletes when compared to non-athletes [74–76], and greater EF has been found in elite athletes when compared to sports performers with less experience or expertise [27,28,77]. Within adolescence, it has been found that elite soccer players obtained greater EF scores than a standardized norm group of men and women [78]. An approach known as the “cognitive component skills approach” investigates the association between sports expertise and cognitive test performance that are relevant to the cognitive requirements in elite sports [79]. Specifically, this approach investigates cognitive functions including working memory, cognitive flexibility, and inhibition [79]. Some studies failed to align with these results [80,81]. Although elite sports have demonstrated significance in terms of bettering an individual’s EF [28,74,75], this factor was not accounted for in this study as this study placed a focus on non-elite athletes, PA level, and meeting the PA guidelines. Therefore, participants were not questioned whether they participated in sport at an elite level, and thus the potential association between PA and EF may still exist if the confounding variable was included.”

5

Explain how MET was calculated?

Thank you for this comment. METs were not calculated for this study, but the authors understand the confusion with MET and ‘met PA guideline’. This has now been edited in table 4 and throughout the manuscript to “achieved” and “not achieved” guidelines to avoid future confusion.

6

What issues and questions remain unresolved or emerge from the results in this study?

The authors appreciate this being highlighted. However, the authors believe that all issues and questions that stem from the results of this study have been discussed throughout the manuscript. For the reviewer’s reference we have highlighted areas where these are considered:

This can be found on lines 354-356, which reads “Thus, researchers and health professionals should consider sleep duration as a potential contributor to young adult cognitive functioning. This factor may explain the no significant associations between PA and EF in this study”.

Lines 369-385 discusses the potential factor of circadian arousal as an influence upon individual’s EF. Please see lines 377-382, which reads “This has been explained through circadian arousal changes that stem from body temperature adaptations throughout the day, namely an increase as the day progresses, which is said to promote optimum performance on complex cognitive processes, such as working memory [64]. Given that participants completed cognitive testing at different times throughout the day, the literature is suggestive of potential inaccuracies within the EF data of this study”, which highlights the question of whether circadian arousal may impact EF.

Moreover, genetic variation has also been questioned as potential factor in participant’s cognitive abilities. This can be found in lines 386-397.

The issue of sedentary behaviour was also raised, as lines 399-405 highlights “Considering this study focused on achieving PA guidelines, individuals who did not meet the PA guideline involved sedentary participants and those who engaged in little PA that failed to meet the guideline. This is noteworthy as participants in the ‘not achieved’ group may have engaged in SB, potentially negatively impacting their EF scores. Thus, further analysis is warranted to explore this potential association between SB and EF in adolescents”.

The issue of executive dysfunction within those with autism was discussed for future research opportunities, please see lines 449-460.